# Musculoskeletal Diseases Role in the Frailty Syndrome: A Case–Control Study

**DOI:** 10.3390/ijerph191911897

**Published:** 2022-09-20

**Authors:** Francesco Cattaneo, Ilaria Buondonno, Debora Cravero, Francesca Sassi, Patrizia D’Amelio

**Affiliations:** 1Department of Public Health Sciences and Pediatrics, University of Torino, 10126 Torino, Italy; 2Geriatric and Bone Disease Unit, Department of Internal Medicine, University of Torino, 10126 Torino, Italy; 3Service of Geriatric Medicine & Geriatric Rehabilitation, Department of Medicine, University of Lausanne Hospital (CHUV), 1011 Lausanne, Switzerland

**Keywords:** frailty, osteoporosis, sarcopenia, muscle, bone, fractures

## Abstract

Frailty syndrome severely burdens older age, and musculoskeletal diseases are of paramount importance in its development. The aim of this study is to unravel the contribution of musculoskeletal diseases to frailty syndrome. This is a case–control study, and we enrolled 55 robust community-dwelling age- and gender-matched patients, with 58 frail and pre-frail subjects. Frailty was diagnosed according to the Fried criteria (FP), and the Fragility Index (FI) was calculated. In all the subjects, a comprehensive geriatric assessment was carried out. Their nutritional status was evaluated by the Mini Nutritional Assessment and Bioelectrical Impedance Analyses. Their bone density (BMD), bone turnover, muscle mass, strength and performance were evaluated. Here, we show that the prevalence of frailty varies according to the diagnostic criteria used and that FP and FI showed a moderate to good agreement. Despite age and gender matching, frail subjects had lower muscle strength, performance and BMD. Their quality of life and cognitive performance were reduced in the frail subjects compared to the robust ones. Muscular strength and performance, together with mood, significantly predicted the diagnosis of frailty, whereas BMD and bone turnover did not. In conclusion, we show that sarcopenia plays a pivotal role in predicting the diagnosis of frailty, whereas osteoporosis does not.

## 1. Introduction

During the last century, we have witnessed a progressive increase in life expectancy that is not comparable to the increase in healthy lifespans; hence, unhealthy aging represents an enormous burden for the individual, the society and the healthcare system [1]. Thus, defining the characteristics leading to frail or robust aging is of paramount importance. Frailty is a clinical syndrome with multiple components associated with a reduced homeostasis, increased risk of poor health outcomes, such as loss of independence, the presence of multiple diseases, increased rates and lengths of hospitalization, falls, fall-related injuries, mortality [2] and, consequently, increased health-related costs [3]. Although several definitions of frailty have been proposed, two models are the most diffused and used in the literature and have been extensively validated; these are the Fried frailty phenotype (FP) [4] and the Frailty Index of the accumulation of deficits (FI) [5,6].

The FP is mostly centered on physical frailty and takes into great account the physical performance and muscular force; according to this model, the subjects are classified as robust, pre-frail or frail, depending on the presence of one or more criteria [4]. The use of the FI takes into account the accumulation of several factors both physical and psychosocial in order to identify a frail phenotype [5], and frailty is defined as a proportion of deficits present in an individual and mathematically derived. Differently from the FP, the FI considers different the dimensions of frailty: the physical capacity, the cognition and the quality of life. The two definitions appears to have a modest correlation [7]; hence, the prevalence of the frailty syndrome significantly varies according to the different diagnostic criteria [8].

The development of the frailty syndrome is multifactorial: physical, psychological and social factors play an important role in its pathogenesis [9]. The early identification of the factors leading to frailty is important to accelerate the diagnosis and treatment of this syndrome, leading to improved health outcomes and quality of life of older adults. Amongst the factors contributing to frailty, the role of musculoskeletal diseases and the associated decline in physical performance appear to be of paramount importance. Musculoskeletal diseases increase in association with aging, and, in particular, the loss of muscle strength, performance and mass, defined as sarcopenia, and loss of bone density and quality, defined as osteoporosis, are highly prevalent in older ages and significantly contribute to unhealthy aging [10,11,12,13]. These two conditions are frequently associated with the framework of the so-called “osteosarcopenia” syndrome [14].

Even though sarcopenia and osteoporosis severely affect healthy aging, their contributions to the frailty syndrome are not completely clear. In particular, the presence of confounding factors such as age and gender, which significantly influence the prevalence of all these conditions, complicates the clear understanding of the role of musculoskeletal diseases in the development and diagnosis of frailty syndrome [15].

Moreover, the criteria used to define frailty are likely important in defining the impact of musculoskeletal health in this syndrome; it is evident that using a physical phenotype of frailty may influence our perception of the role of these diseases and, in particular, of sarcopenia in contributing to frailty.

Furthermore, reduced mobility due to sarcopenia and to osteoporotic fractures may play an important role in reducing the quality of life in frail patients. Although frailty, per se, significantly affects the perceived quality of life in older adults [16], a proper mobility is associated with the quality of life [17], and a recent paper showed that reduced mobility due to the COVID-19 pandemic restrictions had a higher impact on frail compared to non-frail older adults [18]. In particular, musculoskeletal diseases increase the risk of falls [19], and falls and fear of falling greatly affect older patients’ quality of life [20].

Unraveling the contribution of musculoskeletal health in the transition from a robust to frail phenotype using a comprehensive definition of frailty appears to be important in order to diagnose this transition early and to target health interventions. Thus, the aim of this study was to investigate the associations between sarcopenia, osteoporosis and frailty, defined using both the FP and the FI, and to quantify the contributions of these diseases to frailty in older community-dwelling subjects frail or robust and matched for age and gender.

Here, we show that sarcopenia plays an important role in the definition of frailty, and worsens the frail patients’ clinical phenotype, whereas osteoporosis does not.

## 2. Materials and Methods

This is a case–control study; we enrolled in the study 103 community-dwelling subjects coming to the outpatient service of the Geriatric and Bone Disease Unit of the University of Torino (Italy) or to their general practitioner. Subjects were recruited according to the presence or absence of frailty defined according to the Fried criteria; non-frail patients were considered as controls age- and gender-matched (1:1).

Subjects were classified as frail if they had three or more criteria and pre-frail if they had at least one according to the FP definitions [4]. The five criteria comprised unintentional weight loss of at least 4.5 kg during the last year, weakness auto-referred for at least three days/week, low physical activity measured with the Physical Activity Scale for the Elderly (PASE), reduced strength measured with a hand dynamometer and slowness measured with the 4-m walking test.

The study protocol was approved by the ethical committee of the “Città della salute e della Scienza (Torino)” on 23 May 2018, protocol number 0053124. The first patient was enrolled on 2 July 2018, while the last was enrolled on 11 November 2019. Frail subjects and matched controls were enrolled in the same year period. 

Inclusion criteria were age ≥75 years and willing to participate in the study.

Exclusion criteria were a diagnosis of neurocognitive disorders, presence of cancer, presence of renal failure and use of drugs active on bone metabolism, such as an antiresorptive, teriparatide, corticosteroids or immunosuppressants.

Two in-training geriatricians supervised by an experienced geriatrician carried out all the clinical tests. Before the beginning of the study, the in-training geriatricians were specifically formed in order to perform all the described evaluations.

### 2.1. Clinical Evaluation

Frailty was evaluated as both a categorical and a continuous variable using the FP and the FI. Subjects were classified as robust, pre-frail or frail according to the FP criteria, and the FI was calculated as previously described [5,6].

Each patient benefited from a comprehensive geriatric assessment; the following data were recorded: performance in basic activities of daily living (ADL) [21], in instrumental activities of daily living (IADL) [22], the Mini Mental State Examination (MMSE) [23], Geriatric Depression Scale 30 Items (GDS-30) [24] and Cumulative Illness Rating Scale (CIRS) [25].

The nutritional status was evaluated by the Mini Nutritional Assessment (MNA) 30-items questionnaire [26] and by Bioelectric Impedance Analyses (BIA 101, Akern, Florence, Italy). The MNA test provides a full and rapid assessment of the nutritional risk in old subjects; it comprehensively measures weight loss, the Body Mass Index (BMI) and anthropometric measures, the assessment of lifestyle, medication, mobility, a dietary assessment and a subject self-assessment. The MNA (total score 30 points) defines no risk of undernutrition (score ≥ 24), those at risk of undernutrition (score 17 to 24) and those with undernutrition (score < 17) [26].

The use of the BIA allows us to calculate the muscle mass, fat mass and phase angle. The phase angle is calculated from resistance (R) and reactance (Xc) as the arc tangent (Xc/R) × 180°/π and is useful for the diagnosis of malnutrition and clinical prognosis for different diseases [27].

### 2.2. Quality of Life 

In order to measure patients’ quality of life in relation with health, we used the 36-Item Short-Form Health Survey questionnaire (SF-36) [28]. The SF-36 comprehensively assesses the quality of life by taking into account eight domains (physical function and role, body pain, general health, vitality, social functioning, emotional role and mental health).

### 2.3. Muscle Health

The appendicular skeletal muscle mass (ASMM) was measured by the BIA analyses thanks to the use of the equation developed by Sergi and colleagues [29]. Muscular strength was evaluated by measuring their handgrip strength using a Jamar hand dynamometer (MSD, Europe) according to a previous work [30]. Muscular performance was measured using the Short Physical Performance Battery (SPPB) [31] and the Time Up and Go (TUG) test [32].

The SPPB comprehensively assesses the physical performance, evaluating balance, strength and gait speed, and is composed of three tests: standing static balance, lower limb strength measured through the chair stand test and the 4-m walking speed. A SPPB score lower than or equal to 8 points identifies a decline in physical performance [30]. The TUG test assesses the patient’s mobility and balance and measures the time taken by the patient to rise from a chair, walk three meters, walk back and sit down in the chair. A TUG longer that 10 s is common amongst frail older subjects. A TUG longer than 20 s means that the person needs assistance, and a TUG equal to or higher than 30 s indicates an increased risk of falls [32].

In order to further evaluate the risk of falls, we used the Tinetti Gait and Balance Instrument; briefly, the patient sits in a hard, armless chair and is asked to rise and stay standing, then turn 360° and sit back down. Next, the patient walks at a normal speed, turns, walks back and sits down. The tester scores the patient’s performance: the higher the score, the better the performance. The maximum total score is 28 points. A Tinetti lower than 19 identifies patients at a high risk of falling. 

Sarcopenia was diagnosed according to the EWSGOP-2 criteria [30].

### 2.4. Bone Health

Bone mineral density (BMD) was measured at the hip and spine by means of a Hologic QDR 4500 X-ray densitometer; the history of fragility fractures was recorded, and the presence of a vertebral fracture was evaluated using DXA morphometry and expressed as the Spinal Deformity Index (SDI), which allows us to record both the number and severity of vertebral fractures. The calcium, phosphate, PTH and 25OH vitamin D levels were measured with the O-cresolftaleina method, enzymatic method, Chemiluminescent Microparticle Immunoassay (CMIA) and Immunochemiluminometric assay (ICMA), respectively. Bone turnover markers were measured by ELISA in patients’ sera by assessing TRAP5b as an index of bone resorption (Quidel, San Diego, CA, USA), P1NP (Mybiosource, San Diego, CA, USA) and osteocalcin (Mybiosource, San Diego, CA, USA).

Patients were classified as osteoporotic if their T-score was equal to or lower than −2.5 SD, according to the WHO criteria [33].

### 2.5. Statistical Analyses

Frail, pre-frail and robust subjects, as well as sarcopenic and non-sarcopenic ones, were compared for continuous Gaussian variables by means of one-way ANOVA and for categorical variables by means of the χ^2^ test. Diagnosis of frailty by the FI was compared with the FP by means of the Receiver Operator Curve (ROC) analyses; the Area Under the ROC (AUROC), the sensitivity (SE) and the specificity (SP) of the FI were calculated. The intra-test agreement was calculated by the Cohen Index.

In order to evaluate the role of musculoskeletal characteristics in predicting the diagnosis of frailty according to the FI, we built two linear regression models; in the second, the variables nonsignificant in the first model were removed. SPSS 25.0 was used for the statistical analyses, and *p* < 0.05 was considered statistically significant.

## 3. Results

We recruited 48 subjects pre-frail (13, 27%) or frail (35, 73%) according to the FP definition; in parallel, we recruited 55 robust subjects age- and gender-matched with the frail and pre-frail subjects.

Frail and pre-frail subjects had the worst nutritional status and were more dependent, with a higher Comorbidity Index. The physical performance was impaired in frail patients, as expected, and cognitive performance, mood and quality of life were significantly reduced in the frail and pre-frail subjects compared to the robust subjects; these characteristics were also significantly different between the frail and pre-frail subjects. Table 1 shows the general characteristics of the three groups.

The prevalence of malnutrition and risk of malnutrition evaluated by the MNA were significantly different amongst the three groups. The MNA score was lower than 17, indicating malnutrition in 11 frail subjects (31.4%) and in 2 (3.6%) robust subjects, whereas it was between 17 and 24 (at risk of malnutrition) in 14 (40%) frail subjects, 1 (7.7%) pre-frail subject and 5 (9.1%) robust subjects (χ^2^ = 0.54, *p* < 0.0001).

Differently from the FP, the use of the FI classified 18 subjects (17.5%) as robust, 64 (62.1%) as pre-frail and 21 (20.4%) as frail.

Considering the FP as the gold standard, for the threshold 0.25 indicated for the diagnosis of frailty, the FI showed a SE of 100% and a SP of 26.5%, with a very good AUROC of 0.971 ± 0.013 (CI 0.946–0.997, *p* < 0.0001). For the pre-frail subjects (threshold 0.25–0.08), the FI showed a SE varying from 92.3% for the highest cut-off to 0 for the lowest and a SP varying from 18.9% for the highest cut-off to 72.2%, with a poor AUROC of 0.442 ± 0.062 (CI 0.321–0.563, *p* = 0.348) (Figure 1).

The two tests moderately agreed (43.3%, K Cohen = 0.116) on classifying pre-frail subjects and had a good agreement (69.8%, K Cohen = 0.434) on classifying frail subjects.

### 3.1. Frailty and Quality of Life

All eight scales assessed by the SF-36 were significantly reduced in the frail subjects; moreover, physical and social functioning, vitality and mental and general health were significantly more affected in frail than in pre-frail subjects (Table 2).

### 3.2. Frailty and Muscle Heath

Muscle strength and performance, but not muscle mass, were significantly reduced in the frail and pre-fail subjects (Table 3). Nineteen subjects were affected by sarcopenia according to the EWSGOP2 criteria [30]: all of them were classified as pre-frail or frail according to the FP, and sarcopenia was not diagnosed in the robust subjects (χ^2^ = 48.9, *p* < 0.0001).

Amongst the frail patients, the diagnosis of sarcopenia was associated with a poorer nutritional status, reduced femoral BMD and higher risk of falls, whereas there were no differences in the prevalence of fractures, calcium, phosphate, 25OHvitamin D and bone turnover markers (Table 4). As regards the quality of life, we found no differences between sarcopenic and non-sarcopenic frail subjects.

### 3.3. Frailty and Bone Health

In order to evaluate bone health in pre-frail, frail and robust subjects, we measured the BMD at the femur and spine, SDI, calcium, phosphate, 25OHvitamin D and bone turnover. Our data showed that, although the diagnosis of osteoporosis according to the WHO criteria was not significantly different amongst the frail, pre-frail (36.7%) and robust subjects (28%, χ^2^ = 0.713, *p* = 0.273), femoral BMD, calcium and phosphate were significantly lower in the frail population. Despite the differences in calcium–phosphate metabolism and BMD, we did not find any significant differences in the bone turnover markers or in the SDI (Table 5).

Seven subjects in the robust (13%), three (23%) in the pre-frail and seven (20%) in the frail group were supplemented with calcium and vitamin D (χ^2^ = 1.287, *p* = 0.525).

In order to evaluate which of the analyzed variables was the most predictive for frailty, we built up two linear regression models: for the first model, we accounted for variables significantly different amongst the robust and frail subjects using the FI as continuous variables for frailty. The second model was obtained after removing variables that were not significant after running the first model. According to these analyses, muscular strength and performance, together with mood, appeared to be the best predictors for frailty syndrome (Table 6).

## 4. Discussion

In this study, we evaluated the clinical features characterizing pre-frail, frail and robust community-dwelling older subjects matched for gender and age with particular regard to the role of the musculoskeletal features in predicting the diagnosis of frailty. As age and gender significantly influence the diagnosis of frailty, the subjects were matched for these variables; nevertheless, frail and pre-frail subjects were significantly different from robust ones in terms of independency, nutritional status, cognitive function and muscular strength and performance, as expected.

The main aim of this study was to evaluate the musculoskeletal health in frailty; here, we show that the muscle strength and performance were significantly decreased in frail subjects compared to the robust ones, whereas, in the pre-frail subjects, only muscle strength and not muscle performance was affected. Interestingly, the muscle mass was not significantly different between frail and non-frail subjects, and this observation may explain the wide difference in the prevalence of sarcopenia observed across several studies according to the different criteria uses. Diagnostic criteria based only on muscle mass and muscle strength overestimate the prevalence of sarcopenia [34].

In our study, all the subjects diagnosed with sarcopenia were also frail or pre-frail; however, not all the frail subjects were diagnosed with sarcopenia. The presence of sarcopenia was associated with a more severe frailty phenotype; sarcopenic subjects display the worst nutritional status and a lower bone density at the femoral neck. Muscular health appears to be an important predictor of frailty, as both muscular strength and performance were predictive of a higher FI. Thus, the authors suggest screening frail patients for sarcopenia in order to propose a targeted intervention by physical exercise and proper nutritional intervention [35].

Contrary to other studies [17,18,36], we did not find significant differences in the quality of life in sarcopenic and non-sarcopenic subjects; this discrepancy may be due to our focus on frailty that negatively impacts on the quality of life per se [16]. Moreover, differently from other studies, we did not measure the quality of life specifically due to mobility [17,18,36], but rather the general health-related quality of life [28].

Several studies have shown that frailty increases the risk of osteoporotic fractures independently of age in older populations (for a review, see [37]). However, the role of osteoporosis in the pathogenesis of frailty is still not clear. Recently, Tembo and coll. [15] suggested that muscular mass and strength, but not bone quality, are predictive of frailty in both men and women. In this study, we showed a significant decrease in femoral bone density in frail and pre-frail subjects compared to robust ones, but no difference in the prevalence and severity of vertebral fractures. Frail patients displayed lower calcium and phosphate levels, suggesting a role of malnutrition in the development of bone loss; however, bone turnover markers or vitamin D levels were not significantly different in the frail subjects. Hypovitaminosis D is highly prevalent amongst older adults, both hospitalized [38] and community dwelling [39], and has been associated with different clinical characteristics belonging to frailty [40]; in our population, older age may mask a possible association of hypovitaminosis with frailty. In terms of frailty, sarcopenic subjects had lower bone density at the femoral neck, but no other differences in bone turnover and calcium–phosphate metabolism, thus confirming the association between sarcopenia and osteoporosis [14].

In our study, only muscle strength and performance were predictive of the frailty syndrome, whereas no parameter of bone health was predictive. These results agree with the recent paper by Tembo and coll. [15] on a larger population. Differently from Tembo and colleagues, we used a case–control design in order to avoid the biasing effect of age and gender, and we evaluated the role of musculoskeletal health in the diagnosis of frailty by considering frailty as a continuum rather than a categorical variable.

It is well-known that sarcopenia and osteoporosis frequently coexist and that muscle mass and bone density are associated [41,42,43]; however, our data, as well as the Tembo and colleagues data [15], suggest that, in the old and frail subjects, sarcopenia, rather than osteoporosis, plays a causal role in the development of the frailty syndrome.

Concerning the other clinical features analyzed in this study, the nutritional status measured by the MNA was significantly worse in frail patients, whereas the BMI and phase angle were not. It is well-known that the BMI is associated with frailty and mortality in a U-shaped curve [44]; this observation, together with our relative small sample size, may explain the similarities in the BMI between the three groups of subjects.

As regards the phase angle, only a few studies investigated this parameter in frailty [45,46,47], suggesting an association with frailty and a correlation with mortality. However, the population included in the above-mentioned studies is highly heterogeneous and noncomparable to the population included in our study; moreover, the criteria used for the diagnosis of frailty were different amongst the different studies. Tanaka and coll. [47] used the Japanese version of the Cardiovascular Health Study criteria [48], and Wilhelm-Leen and coll. [46] and Mullie and coll. [45] used the FP. As regards the population included, Mullie and coll. [45] analyzed patients undergoing cardiac surgery, and Tanaka and coll. [47] evaluated subjects younger than 75 years; hence, their population was not comparable with the one enrolled in this study. Wilhelm-Leen and coll. [46] analyzed a large cohort of community-dwelling subjects and suggested an association between frailty, phase angle and increased mortality. These conflicting results merit deeper evaluations by further ad hoc studies.

Although malnutrition is frequent amongst hospitalized and institutionalized patients, it is also an important feature in community-dwelling older adults; nevertheless, in this population, the diagnosis of malnutrition is frequently neglected [49,50]. The underdiagnosis led to undertreatment and, consequently, to increased negative outcomes of this condition, such as functional decline [51] and increased mortality [52]. Here, we showed that, in the older community-dwelling population, the use of the MNA as a screening tool for malnutrition rather than the BMI and body composition was able to differentiate between frail and non-frail subjects. Moreover, our data confirmed the high prevalence of malnutrition and risk of malnutrition, especially amongst frail subjects, confirming the importance of nutritional screening in this population. According to these data, we suggest evaluating the nutritional status in older community-dwelling subjects as a routine clinical practice in order to identify frail subjects early and suggest targeted nutritional support.

Our data confirmed the need for a broader definition of the frailty syndrome, as we found not only physical, but also psychological and cognitive differences between the three groups of subjects [9]. Frail and pre-frail subjects were significantly less performant in terms of cognition compared to the robust subjects, who reported a better mood and quality of life compared to the frail ones. Indeed, the mood, evaluated by the GDS, was significantly predictive of the FI, thus confirming the important role of cognitive and not only physical health in the development of the frailty syndrome. Nevertheless, according to other studies [53], we found a good agreement between the physical definition of frailty (FP) and the FI in detecting frail subjects, whereas the agreement in detecting pre-frail subjects was only moderate. In our cohort, indeed, the pre-frail subjects were more similar to the robust than to the frail subjects, in terms of the nutritional status, presence of comorbidity and independence. As regards musculoskeletal health, similar to frail subjects, pre-frail subjects had reduced muscular strength, but a muscle mass and performance similar to the robust ones.

The main limit of our study is the relatively small cohort enrolled; the high number of tests performed may interfere with the data analyses, as some variables may be highly influenced by others, such as the BMD, which may be influenced by muscle mass, strength and performance [14,43]. Thus, analyses conducted in a larger sample size may reveal the direct effects of BMD on the diagnosis of frailty, independently from the muscle mass, strength and performance. Despite this main limit, our results were in line with similar studies performed in larger cohorts [15].

One of the main strengths of this study is the enrollment of controls age- and gender-matched with frail subjects; this matching allowed us to reduce the bias due to the interference of age and gender on the different variables analyzed. Other strengths are the enrollment of the “oldest old” subjects [54] and the complete characterization of all the subjects involved by a geriatric assessment, measurement of the muscle mass, strength and performance, bone density and turnover. Furthermore, here, we applied two of the main indices used for the identification of frail subjects, the FP and FI, and the use of the FI allowed us to evaluate the effect of musculoskeletal diseases on frailty as a continuum and not only as categorical variables.

## 5. Conclusions

Here, we confirmed that frailty is the result of physical, cognitive and quality of life impairment. Malnutrition is significantly prevalent amongst frail community-dwelling subjects. Sarcopenia significantly worsened the frail phenotype and had a role in the diagnosis of frailty, even according to the FI. Osteoporosis and bone metabolism did not play a significant role in predicting frailty.

Hence, we suggest including in the clinical routine evaluation screening for sarcopenia and malnutrition in order to identify frail patients early and to treat them with a proper nutritional approach combined with physical exercise. According to previous studies, this combined approach is useful not only in the treatment of sarcopenia and frailty [35,55,56] but also efficiently improves patients’ quality of life [57].

## Figures and Tables

**Figure 1 ijerph-19-11897-f001:**
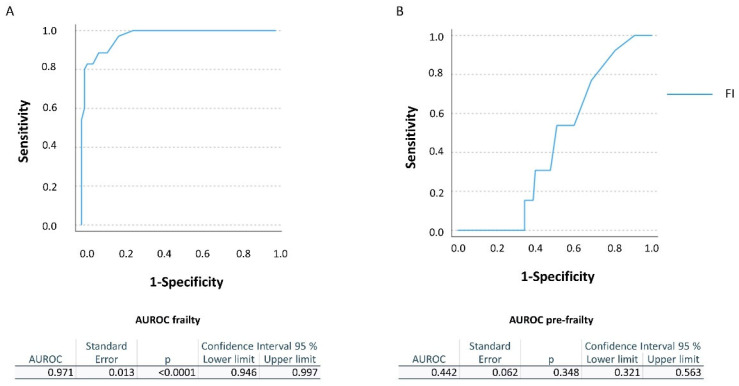
**ROC curves for the FP and FI**. (**A**) The AUROC for the FI in diagnosing frailty compared to the FP. (**B**) The AUROC for the FI in diagnosing pre-frailty compared to the FP.

**Table 1 ijerph-19-11897-t001:** **General characteristics of the subjects classified according to the FP criteria**. The mean and SD with CI are shown, and the *p*-values were calculated with one-way ANOVA.

		Mean ± SD (95% Confidence Interval)	*p* Over All	*p* Frail vs. Pre-Frail
**FI (score)**	*Robust (55)*	0.1 ± 0.04 (0.09–0.11)	<0.001	0.018
	*Pre-frail (13)*	0.1 ± 0.04 (0.12–0.16)		
	*Frail (35)*	0.3 ± 0.1 (0.27–0.35)		
**Age (y)**	*Robust (55)*	81 ± 6 (79.8–83)	0.160	0.511
	*Pre-frail (13)*	79 ± 4 (76.3–81)		
	*Frail (35)*	80 ± 5 (77.9–81.5)		
**Drugs/daily**	*Robust (55)*	4 ± 2 (3–5)	0.024	0.671
	*Pre-frail (13)*	3 ± 2 (2–5)		
	*Frail (35)*	5 ± 2 (4–6)		
**BMI**	*Robust (55)*	25.6 ± 3.5 (24.7–26.6)	0.090	0.255
	*Pre-frail (13)*	25.5 ± 4.7 (22.7–28.4)		
	*Frail (35)*	23.7 ± 5.1 (21.9–24.4)		
**Fat mass (%)**	*Robust (55)*	25 ± 4.6 (23.8–26.3)	0.049	0.209
	*Pre-frail (13)*	24.7 ± 4.4 (22.1 ± 27.4)		
	*Frail (35)*	22.3 ± 6.4 (20.0 ± 24.5)		
**Phase (°)**	*Robust (55)*	5.1 ± 1 (4.8–5.4)	0.185	0.335
	*Pre-frail (13)*	5.7 ± 1(5.1–6.3)		
	*Frail (35)*	5 ± 1.3 (4.5–5.5)		
**MNA (score/30)**	*Robust (55)*	26 ± 3.8 (25–27)	<0.0001	<0.0001
	*Pre-frail (13)*	26 ± 2.3 (25–28)		
	*Frail (35)*	19.4 ± 5.6 (17–21)		
**MMSE (score/30)**	*Robust (55)*	28 ±1 (27–28)	<0.0001	0.600
	*Pre-frail (13)*	27 ± 2 (26–28)		
	*Frail (35)*	27 ± 12 (26–27)		
**CIRS (score/30)**	*Robust (55)*	9.3 ± 4.0 (8.2–10.4)	<0.0001	0.013
	*Pre-frail (13)*	10.8 ± 12.4 (9.4–12.2)		
	*Frail (35)*	13.7 ± 15.5 (11.8–15.7)		
**GDS (score/30)**	*Robust (55)*	7.8 ± 5.1 (6.5–9.2)	<0.0001	0.007
	*Pre-frail (13)*	8.1 ± 4.3 (5.5–10.7)		
	*Frail (35)*	17.4 ± 7.6 (14.8–20)		
**ADL** **(number of lost function)**	*Robust (55)*	0.01 ± 0.19 (0–0.9)	<0.0001	<0.0001
	*Pre-frail (13)*	0 ± 0 (0–0)		
	*Frail (35)*	1.2 ± 1.9 (0.5–1.9)		
**IADL (score/14)**	*Robust (55)*	12.5 ± 2.6 (11.8–13.2))	<0.0001	<0.0001
	*Pre-frail (13)*	13.5 ± 0.2 (13.1–13.9)		
	*Frail (35)*	8.2 ± 4.5 (6.6–9.8)		

**Table 2 ijerph-19-11897-t002:** **Quality of life of the subjects classified according to the FP criteria**. Mean and SD with CI are shown, and the *p*-values were calculated with one-way ANOVA.

		Mean ± SD (95% Confidence Interval)	*p* Over All	*p* Frail vs. Pre-Frail
**Physical functioning**	*Robust (55)*	84.4 ± 11 (81.4–87.3)	<0.0001	0.043
	*Pre-frail (13)*	73.9 ± 15.4 (64.5–83.2)		
	*Frail (35)*	41.5 ± 28 (31.7–51.2)		
**Role physical**	*Robust (55)*	73.9 ± 27.5 (66.5–81.3)	<0.0001	0.304
	*Pre-frail (13)*	76.9 ± 27.9 (60.1–93.8)		
	*Frail (35)*	35.2 ± 32.0 (24.0–46.3)		
**Role emotional**	*Robust (55)*	83.5 ± 21.3 (77.7–89.2)	<0.0001	0.186
	*Pre-frail (13)*	69.2 ± 23.3 (53.9–84.5)		
	*Frail (35)*	37.2 ± 32.6 (25.8–48.6)		
**Vitality**	*Robust (55)*	71.7 ± 11.2 (68.3–74.7)	<0.0001	0.021
	*Pre-frail (13)*	62.3 ± 12.0 (55.1–69.6)		
	*Frail (35)*	39.4 ± 20.7 (32.2–46.6)		
**Mental health**	*Robust (55)*	77.0 ± 13.0 (73.5–80.6)	<0.0001	<0.0001
	*Pre-frail (13)*	74.8 ± 15.7 (65.3–84.3)		
	*Frail (35)*	52.8 ± 21.1 (45.5–60.2)		
**Social functioning**	*Robust (55)*	85.5 ± 15.0 (81.4–89.5)	<0.0001	<0.0001
	*Pre-frail (13)*	89.4 ± 6.9 (85.2–93.6)		
	*Frail (35)*	50.0 ± 33.0 (38.5–61.5)		
**Bodily pain**	*Robust (55)*	76.9 ± 21.1 (71.2–82.6)	<0.0001	0.514
	*Pre-frail (13)*	76.2 ± 24.8 (61.2–91.1)		
	*Frail (35)*	48.3 ± 27.1 (38.8–57.2)		
**General health**	*Robust (55)*	70.5 ± 13.9 (66.7–74.2)	<0.0001	0.002
	*Pre-frail (13)*	62.7 ± 10.1 (56.6–68.8)		
	*Frail (35)*	41.3 ± 25.9 (32.3–50.4)		

**Table 3 ijerph-19-11897-t003:** **Muscle health amongst the subjects classified according to the FP criteria**. Mean and SD with CI are shown, and the *p*-values were calculated with one-way ANOVA.

		Mean ± SD (95% Confidence Interval)	*p* Overall	*p* Frail vs. Pre-Frail
**Hand grip strength (Kg)**	*Robust (55)*	30.6 ± 1.1 (28.4–32.7)	<0.0001	0.695
	*Pre-frail (13)*	23.9 ± 1.4 (21.1–26.8)		
	*Frail (35)*	16.8 ± 5.3 (14.9–18.6)		
**ASMM (Kg/m^2^)**	*Robust (55)*	8.3 ± 3.6 (7.3–9.2)	0.855	0.800
	*Pre-frail (13)*	7.7 ± 1.3 (6.8 ± 8.5)		
	*Frail (35)*	8.0 ± 4.0 (6.6–9.4)		
**SPPB (score/30)**	*Robust (55)*	9 ± 1.6 (8.5–9.4)	<0.0001	<0.0001
	*Pre-frail (13)*	8 ± 1.2 (7.4–8.8)		
	*Frail (35)*	3.3 ± 2.5 (2.5–4.2)		
**TUG (sec)**	*Robust (55)*	11 ± 3 (10.3–12.0)	<0.0001	<0.0001
	*Pre-frail (13)*	11 ± 1 (10.3–11.8)		
	*Frail (35)*	20 ± 9 (17–23.5)		
**Tinetti (score/28)**	*Robust (55)*	26 ± 3 (25–26.8)	<0.0001	<0.0001
	*Pre-frail (13)*	24 ± 3 (22.5–26.3)		
	*Frail (35)*	15 ± 7 (12.5–17.5)		

**Table 4 ijerph-19-11897-t004:** **Clinical characteristics of frail and pre-frail patients with or without sarcopenia**. Mean and SD with CI are shown, and the *p*-values were calculated with one-way ANOVA.

Variable	Sarcopenic Frail or Pre-Frail (19)	Non Sarcopenic Frail or Pre-Frail (29)	*p*-Value
**Age**	81 ± 3 (79–82)	78 ± 6 (76–80)	0.059
**Gender**	42%(F) 58%(M)	45% (F) 55% (M)	0.556
**FI**	0.28 ± 0.1 (0.2–0.3)	0.22 ± 0.1 (0.2–0.3)	0.125
**BMI**	22.1 ± 5 (19.7–24.5)	25.5 ± 4.5 (23.8–27.3)	0.018
**MNA**	18.7 ± 5.6 (16.0–21.4)	22.9 ± 5.4 (20.9–24.9)	0.012
**Phase (°)**	4.4± 1.0(3.7–5.2)	7.4 ± 1.2 (5.6–9.2)	0.023
**Numbers of drugs assumed daily**	5 ± 3 (3.6–6.4)	4.5 ± 2.2 (3.7–5.4)	0.591
**CIRS**	13.8 ± 5.0 (11.3–16.3)	12.4 ± 5.0 (10.5–14.3)	0.356
**ADL (number of lost functions)**	1 ± 1.7 (0.15–1.85)	0.88 ± 1.7 (0.38–1.37)	0.618
**IADL (number of functions)**	8 ± 4 (6–10)	11 ± 4 (9–12)	0.051
**GDS-30**	13.4 ± 7.8 (10.4–16.4)	17.2 ± 7.8 (13.4–21.0)	0.109
**MMSE**	26.4 ± 1.6 (26.3–27.2)	27.0 ± 1.5 (26.4–27.6)	0.188
**Tinetti**	13.4 ± 6.4 (10.3–16.4)	20.3 ± 7.3 (17.5–23.1)	0.002
**TUG**	21.6 ± 10.5 (16.2–27.0)	14.8 ± 5.5 (12.5–17.0)	0.008
**Chair test**	20.4 ± 4.7 (17.0–23.7)	16.5 ± 6.6 (13.4–19.6)	0.106
**BMD lumbar spine (gr/cm^2^)**	0.856 ± 0.196 (0.758–0.953)	0.972 ± 0.238 (0.882–1.06)	0.088
**BMD femoral neck(gr/cm^2^)**	0.173 ± 0.129 (0.772–0.887)	0.830 ± 0.148 (0.772–0.887)	0.009
**SDI**	0.78 ± 0.3 (0.2–1.36)	1.1 ± 0.8 (0.05–2.0)	0.444
**Calcium (mMol/L)**	2.2 ± 0.1 (2.1–2.3)	2.3 ± 0.2 (2.2–2.3)	0.127
**Phosphate (mMol/L)**	0.9 ± 0.2 (0.8–1.0)	1.0 ± 0.2 (0.9–1.1)	0.462
**PTH (pg/mL)**	42.9 ± 15.7 (35.1–50.68)	58.2 ± 37.6 (43.9–73.0)	0.108
**25OHvitaminD**	16.2 ± 6.9 (12.8–19.7)	21.3 ± 12.7 (16.4–26.1)	0.129
**P1NP (pg/mL)**	360.9 ± 224.9 (241.1–480.8)	443.9 ± 223.6 (342.1–545.6)	0.272
**OC (ng/mL)**	14.2 ± 9.8 (9.0 ± 19.4)	13.8 ± 7.2 (10.5–17.0)	0.869
**TRAP5b (UI/mL)**	10.1 ± 3.7 (8.9–11.4)	10.3 ± 3.6 (8.7–11.9)	0.862

**Table 5 ijerph-19-11897-t005:** **Bone health amongst the subjects classified according to the FP criteria.** Mean and SD with CI are shown, and the *p*-values were calculated with one-way ANOVA.

		Mean ± SD (95% Confidence Interval)	*p* Overall	*p* Frail vs. Pre Frail
**Calcium (mMol/L)**	*Robust (55)*	2.3 ± 0.1 (2.2–2.3)	0.009	0.011
	*Pre-frail (13)*	2.4 ± 0.1 (2.3–2.4)		
	*Frail (35)*	2.2 ± 0.2 (2.2–2.3)		
**Phosphate (mMol/L)**	*Robust (55)*	1.0 ± 0.1 (0.9–1.0)	0.020	0.018
	*Pre-frail (13)*	1.1 ± 0.1 (1.0–1.1)		
	*Frail (35)*	0.9 ± 0.2 (0.8–1.0)		
**PTH (pg/mL)**	*Robust (55)*	55.2 ± 24.0 (48.7–61.7)	0.565	0.414
	*Pre-frail (13)*	46.2 ± 14.4 (37.4–54.9)		
	*Frail (35)*	54.7 ± 36.1 (42.1–67.3)		
**TRAP5b (IU/mL)**	*Robust (55)*	9.5 ± 3.5 (8.4–10.6)	0.422	0.470
	*Pre-frail (13)*	8.9 ± 4.5 (4.8–13.0)		
	*Frail (35)*	10.5 ± 3.6 (9.2–11.8)		
**P1NP pg/mL**	*Robust (55)*	477.3 ± 192.9 (416.4–538.2)	0.359	0.614
	*Pre-frail (13)*	409.0 ± 263.5 (165.3–652.6)		
	*Frail (35)*	409.0 ± 217.6 (327.8–491.2)		
**OC (ng/mL)**	*Robust (55)*	15.1 ± 6.1 (13.1–17.0)	0.440	0.284
	*Pre-frail (13)*	16.6 ± 11.3 (6.2–27.1)		
	*Frail (35)*	13.3 ± 7.5 (10.5–16.1)		
**25OHvitaminD (ng/mL)**	*Robust (55)*	17.9 ± 9.3 (15.4–20.5)	0.072	0.631
	*Pre-frail (13)*	24.0 ± 14.3 (15.4–32.7)		
	*Frail (35)*	17.5 ± 9.2 (14.3–20.7)		
**BMD femoral neck (g/cm^2^)**	*Robust (55)*	0.853 ± 0.163 (0.807–0.898)	0.020	0.058
	*Pre-frail (13)*	0.855 ± 0.181 (0.740–0.970)		
	*Frail (35)*	0.759 ± 0.133 (0.712–0.806)		
**BMD lumbar spine (g/cm^2^)**	*Robust (55)*	1.006 ± 0.199 (0.952–1.060)	0.165	0.651
	*Pre-frail (13)*	0.952 ± 0.258 (0.797–1.108)		
	*Frail (35)*	0.918± 0.219 (0.842–0.995)		
**SDI (score)**	*Robust (55)*	0.95 ± 1.5 (0.53–1.36)	0.671	0.690
	*Pre-frail (13)*	1.00 ± 1.73 (0.05–2.05)		
	*Frail (35)*	1.26 ±1.96 (0.58–1.95)		

**Table 6 ijerph-19-11897-t006:** **Linear regression models predictive of frailty**. The standardized beta coefficient, t, *p*-values, 95% CI and partial correlation are shown. (A): linear regression models taking into account variables significantly different amongst fit and frail subjects. (B): linear regression adjusted model.

**A—Model 1: adjusted R square = 0.693, *p* < 0.001**					
**Introduced Variables**	**Standardized Beta**	**t**	** *p* **	**95% Confidence Interval**	**Partial Correlation**
**SPPB (score/30)**	−0.080	−0.701	0.486	−0.010; 0.005	−0.040
**TUG (sec)**	−0.146	−1.285	0.202	−0.005; 0.001	−0.074
**Hand grip strength (Kg)**	0.154	2.044	0.044	0.001; 0.066	0.117
**GDS (score/30)**	0.345	4.392	<0.001	0.003; 0.007	0.252
**MMSE (score/30)**	−0.021	−0.311	0.756	−0.010; 0.007	−0.018
**MNA (score/30)**	−0.097	−0.963	0.338	−0.006; 0.002	−0.055
**Tinetti (score/28)**	−0.496	−4.182	<0.001	−0.012;−0.179	−0.240
**Calcium (mmol/L)**	0.089	1.234	0.221	−0.042;−0.179	0.071
**Phosphate (mmol/L)**	−0.130	−1.694	0.094	−0.165; 0.013	−0.097
**BMD femoral neck (g/cm^2^)**	0.014	0.224	0.823	−0.070; 0.088	0.013
**Fat mass (%)**	0.068	0.922	0.359	−0.001; 0.004	0.053
**B—Model 2: adjusted R square = 0.736, *p* < 0.001**					
**Introduced Variables**	**Standardized Beta**	**t**	** *p* **	**95% Confidence Interval**	**Partial Correlation**
**Hand grip strength (Kg)**	0.149	2.262	0.026	0.004; 0.068	0.222
**GDS (score/30)**	0.361	5.589	<0.001	0.004; 0.008	0.490
**Tinetti (score/28)**	−0.492	−7.149	<0.001	−0.010;−0.006	−0.584

## Data Availability

The data are available upon reasonable request to the corresponding authors.

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
