# Peer review of "Musculoskeletal Diseases Role in the Frailty Syndrome: A Case–Control Study"

_ijerph, 2022, doi:10.3390/ijerph191911897_

Round 1
Reviewer 1 Report
The authors conducted a case controll study with comparison of individuals without frailty, pre-frailty and frailty. The diagnosis of frailty is based on the frailty phenotype by Linda Fried, one of three validated measures to assess frailty.
The authors conducted a lot of statistical tests in a small sample. Such a procedure may result in significance purely by chance. This issue must be discussed.
Since assignement to group is based on fraitly status, differences between several assessment tests are not unexpected. Furthermore, osteoporosis is a silent disease, often discoverd after a fracture. Mobility problems are not a consequence of osteoporosis but a consequence of fractures due to falls and osteoporosis (see line 69).
The pivotal question is, how to treat for frailty? This question should be answered by the authors based on the results found.
What consequences should a physicain draw from these results?
Author Response
The authors thanks the reviewer for his/her useful comments
Rev 1: The authors conducted a lot of statistical tests in a small sample. Such a procedure may result in significance purely by chance. This issue must be discussed.
Authors: this point is now discussed and added as a study limit (lines 379-382)
Rev 1: osteoporosis is a silent disease, often discovered after a fracture. Mobility problems are not a consequence of osteoporosis but a consequence of fractures due to falls and osteoporosis (see line 69).
Authors: corrected thank you
Rev 1: The pivotal question is, how to treat for frailty? This question should be answered by the authors based on the results found. What consequences should a physicain draw from these results?
Authors: these points are now discussed throughout the discussion section
Reviewer 2 Report
Thank you for the opportunity to review this paper. While the topic is of interest in its current form it will require more work before publication. There are a number of areas that require rewriting or clarification. I will comment on these areas section by section.
ABSTRACT:
Abstract is unclear and doesn't summarize the aim, methodology and results clearly. Suggest it is re-written.
Introduction.
The introduction is easy to read, however did not extend existing knowledge on this topic. It should include more update indications regarding the relationship among prescription of physical activity, quality of life and falls in elderly people. Authors reported that reduced mobility due to sarcopenia and osteoporosis may play an important role in reducing quality of life of frail patients. However, they did not focus on the topic. See for example Battaglia, G. et al., A. Walking in Natural Environments as Geriatrician’s Recommendation for Fall Prevention: Preliminary Outcomes from the “Passiata Day” Model. Sustainability. 2020; 12: 2684. Lacroix, A et al.. Effects of Supervised vs. Unsupervised Training Programs on Balance and Muscle Strength in Older Adults: A Systematic Review and Meta-Analysis. Sports Med. 2017, 47, 2341–2361.
Please also include a reference to support the following statement " The development of frailty syndrome is multifactorial: physical, psychological and social factors plays an important role in its pathogenesis ".
After the purpose statement, please provide a hypothesis for what the authors think the results will yield.
Methods
Some important information appears to be presently omitted from the methods section. Further description of the sampling procedure would be helpful for the reader. Please explain better how was selected the sample size and how was the data collected. When were the tests administered? Was the time of year, season, and time of day consistent for all subjects sampled? Further explanation about who collected the data is also necessary here. Have you tested the reliability of your data? If yes, please include the results.
Discussion
In general, the first paragraph of the discussion should at least state which hypotheses were supported. Then the authors should follow with how their results compare with similar data, and what the authors results adds to the literature (different / unique aspects of the data). Several points are made in the discussion, but it is not clear to this reviewer how results from the current study are novel or add to the literature. The authors shortly discuss several possible explanations for the findings. The authors did not discuss how this research may be disseminated into greater practice. Moreover, the limitations and the strengths of this research were not discussed at all.
Author Response
The authors thanks the reviewer for his/her useful comments
Rev2: Abstract is unclear and doesn't summarize the aim, methodology and results clearly. Suggest it is re-written.
Authors: rewritten as suggested
Rev2: The introduction is easy to read, however did not extend existing knowledge on this topic. It should include more update indications regarding the relationship among prescription of physical activity, quality of life and falls in elderly people. Authors reported that reduced mobility due to sarcopenia and osteoporosis may play an important role in reducing quality of life of frail patients. However, they did not focus on the topic. See for example Battaglia, G. et al., A. Walking in Natural Environments as Geriatrician’s Recommendation for Fall Prevention: Preliminary Outcomes from the “Passiata Day” Model. Sustainability. 2020; 12: 2684. Lacroix, A et al.. Effects of Supervised vs. Unsupervised Training Programs on Balance and Muscle Strength in Older Adults: A Systematic Review and Meta-Analysis. Sports Med. 2017, 47, 2341–2361.
Authors: the relationship between mobility, falls and quality of life in frail older patients is now deeply discussed and the suggested references have been added.
Rev 2: Please also include a reference to support the following statement " The development of frailty syndrome is multifactorial: physical, psychological and social factors plays an important role in its pathogenesis ".
Authors: done
Rev 2: After the purpose statement, please provide a hypothesis for what the authors think the results will yield.
Authors: done
Rev. 2. Some important information appears to be presently omitted from the methods section. Further description of the sampling procedure would be helpful for the reader. Please explain better how was selected the sample size and how was the data collected. When were the tests administered? Was the time of year, season, and time of day consistent for all subjects sampled? Further explanation about who collected the data is also necessary here. Have you tested the reliability of your data? If yes, please include the results.
Authors: the requested information have been added in material and methods sections (lines 111-112 and 117-119)
Rev.2. In general, the first paragraph of the discussion should at least state which hypotheses were supported. Then the authors should follow with how their results compare with similar data, and what the authors results adds to the literature (different / unique aspects of the data). Several points are made in the discussion, but it is not clear to this reviewer how results from the current study are novel or add to the literature. The authors shortly discuss several possible explanations for the findings. The authors did not discuss how this research may be disseminated into greater practice. Moreover, the limitations and the strengths of this research were not discussed at all.
Authors: discussion has been reworded according to the reviewer’s suggestions.
Round 2
Reviewer 1 Report
Comments have been taken into account
Reviewer 2 Report
In my opinion the manuscript was improved according to indications. For these reasons It could be accepted.